# Advancing Treatment Strategies: A Comprehensive Review of Drug Delivery Innovations for Chronic Inflammatory Respiratory Diseases

**DOI:** 10.3390/pharmaceutics15082151

**Published:** 2023-08-17

**Authors:** Junming Wang, Pengfei Wang, Yiru Shao, Daikun He

**Affiliations:** 1Center of Emergency and Critical Care Medicine, Jinshan Hospital, Fudan University, Shanghai 201508, China; wangjunming156@126.com (J.W.); 22211270009@m.fudan.edu.cn (P.W.); shaoyiru1983@163.com (Y.S.); 2Research Center for Chemical Injury, Emergency and Critical Medicine of Fudan University, Shanghai 201508, China; 3Key Laboratory of Chemical Injury, Emergency and Critical Medicine of Shanghai Municipal Health Commission, Shanghai 201508, China; 4Department of General Practice, Jinshan Hospital, Fudan University, Shanghai 201508, China; 5Department of General Practice, Zhongshan Hospital, Fudan University, Shanghai 200032, China

**Keywords:** drug delivery, chronic inflammatory respiratory diseases, nanoparticle-based drug delivery systems, inhaled corticosteroids, novel biologicals, gene therapy, personalized medicine

## Abstract

Chronic inflammatory respiratory diseases, such as asthma, chronic obstructive pulmonary disease (COPD), and cystic fibrosis, present ongoing challenges in terms of effective treatment and management. These diseases are characterized by persistent inflammation in the airways, leading to structural changes and compromised lung function. There are several treatments available for them, such as bronchodilators, immunomodulators, and oxygen therapy. However, there are still some shortcomings in the effectiveness and side effects of drugs. To achieve optimal therapeutic outcomes while minimizing systemic side effects, targeted therapies and precise drug delivery systems are crucial to the management of these diseases. This comprehensive review focuses on the role of drug delivery systems in chronic inflammatory respiratory diseases, particularly nanoparticle-based drug delivery systems, inhaled corticosteroids (ICSs), novel biologicals, gene therapy, and personalized medicine. By examining the latest advancements and strategies in these areas, we aim to provide a thorough understanding of the current landscape and future prospects for improving treatment outcomes in these challenging conditions.

## 1. Introduction

Chronic inflammatory respiratory diseases, such as asthma and chronic obstructive pulmonary disease (COPD), affect millions of people worldwide and are a leading cause for the increase in lung disease morbidity and mortality [1]. Asthma, as a heterogeneous clinical syndrome, affects over 300 million people worldwide [2]. COPD, a disease mainly associated with long-term smoking, became the third leading cause of death globally in 2020 [3]. Although there are several existing treatments, limited efficacy, side effects, and individual variability still cannot be ignored [4,5,6]. In recent years, there has been a growing interest in the development of targeted drug delivery systems for the treatment of these diseases [7,8,9]. Nanoparticle-based drug delivery systems, inhaled corticosteroids (ICSs), novel biologicals, gene therapy, and personalized medicine have emerged as promising approaches to deliver drugs more effectively and with fewer side effects.

Currently, the development of new nanoparticle-based drug delivery systems that can target specific cells such as lung epithelial cells and macrophages, while minimizing systemic side effects, have received significant attention [10]. These systems utilize nanoparticles, which are tiny particles ranging from 1 to 100 nanometers in size, to encapsulate and deliver drugs directly to the affected areas of the lungs [11]. By modifying the surface properties of nanoparticles, researchers can enhance their ability to selectively bind to specific cell types in the lungs, thereby improving drug delivery efficiency and reducing off-target effects [12]. Furthermore, nanoparticle-based drug delivery systems can protect the drugs from degradation and enhance their stability, ensuring sustained release and prolonged therapeutic effects [13].

In addition to nanoparticle-based systems, inhaled corticosteroids (ICSs) have long been used as a standard treatment for chronic inflammatory respiratory diseases [14,15]. ICSs work by reducing inflammation in the airways, thus alleviating symptoms and preventing exacerbation. Researchers are also exploring novel biological targets and innovative methods for delivering biologicals to the lungs. Gene therapy approaches, including viral-vector-based delivery systems and CRISPR–Cas9 technology, represent another exciting frontier in the treatment of chronic inflammatory respiratory diseases [16,17]. Moreover, personalized medicine approaches take into account an individual’s unique characteristics, such as genetics, biomarkers, and lifestyle factors, to tailor treatments to their specific needs [8,18]. By utilizing advanced diagnostic tools like genomic sequencing and biomarker analysis, healthcare providers can identify patient subgroups who are more likely to respond to a particular therapy, thus optimizing treatment outcomes [19,20]. However, several challenges remain, including optimizing delivery efficiency, ensuring safety, and addressing ethical considerations.

The purpose of this review is to provide an overview of the current research progress in nanoparticle-based drug delivery systems, ICS, novel biologicals, gene therapy, and personalized medicine for the treatment of chronic inflammatory respiratory diseases. In this review, we examine recent advancements, discuss limitations, and explore future directions for each of these therapeutic approaches.

## 2. Nanoparticle-Based Drug Delivery Systems

The application of nanotechnology continues to provide effective strategies in treating various chronic diseases and improving treatment outcomes. Using nanocarrier systems such as liposomes, micelles, and nanoparticles for pulmonary drug delivery has been proven to be a promising noninvasive treatment strategy for achieving drug deposition and controlled release in the lungs [10] (Figure 1). These systems involve the use of engineered particles with dimensions on the nanometer scale to deliver drugs directly to target cells in the lungs [21]. Nanoparticles have several advantages over conventional drug delivery methods, including improved bioavailability, enhanced targeting, and reduced toxicity [22,23].

Liposomes are spherical vesicles composed of lipid bilayers that can encapsulate both hydrophilic and hydrophobic drugs [24]. The size, surface charge, and lipid composition of lipid nanoparticles (LNPs) can be tailored to enhance drug stability, prolong circulation time, and improve biocompatibility [25]. Furthermore, conjugating small-molecule ligands, peptides [26], or monoclonal antibodies [27] to the surface of an LNP can endow it with targeting ability. For example, folate receptors are often found to be overexpressed on macrophages, which makes folate-coupled LNP a great option for delivering anti-inflammatory drugs [28]. There are many factors that can affect the release of cargo carried by LNPs, including temperature, changes in pH values, enzymes, light, etc. Among them, the mechanism of pH change is the most studied, and can cause LNPs to undergo phase transition and achieve higher membrane permeability [29].

In addition to LNPs, there are also some other nanoparticles that have their own characteristics (Table 1). Micelles are another kind of nanoparticle consisting of amphiphilic molecules that form a core-shell structure [30]. Their great solubility allows them to easily penetrate the increased alveolar fluid barrier present in chronic inflammatory lung diseases. A new kind of stabilized phospholipid nanomicelles (SSMs) can reach deep lung tissue and successfully extend the half-life of budesonide in the lung to 18–20 h [31]. Magnetic nanoparticles (MNPs) developed using the magnetofection technique have wide-ranging applications in the fields of biological research and medicine, including drug and gene therapy, magnetic targeting (such as in cancer therapies), and diagnostic imaging as contrast enhancers [32,33]. A representative example is the superparamagnetic iron oxide nanoparticle (SPION), a type of nanoparticle with special magnetism that can be guided through an external magnetic field to locations within the body [34]. They can accurately transport the drugs coated on their surface, mainly some inflammation-related molecular antibodies like IL4Rα and ST2, to the site of the inflammatory lesion [35,36]. A kind of selective organ targeting (SORT) nanoparticle was designed to release its cargo in a controlled manner; it can target the site of inflammation in the lungs and elsewhere while minimizing exposure of healthy tissue in other parts of the body [37]. This targeted drug delivery approach has the potential to reduce drug toxicity and improve patient outcomes [38]. Recently, a growing number of hybrid nanoparticles (HNPs) have emerged that can simultaneously possess the characteristics of different nanoparticles [39]. This has sparked a trend of exploring different combinations of nanoparticles.

Despite the promise of nanoparticle-based drug delivery, there are still several research challenges that need to be addressed. For example, there is a need to develop nanoparticles with optimal physicochemical properties, such as particle size, surface charge, and stability, to ensure effective drug delivery [46]. Recent research has reported that the structure of mesoporous silica nanoparticles (MSNs) can be well controlled with several parameters such as pH, surfactant, silica precursor, and temperature. For instance, Pan et al. prepared a series of size-controlled MSNs with a range of 25–105 nm by simply changing the amount of the basic catalyst triethanolamine (TEA) added [47]. So, it is believed that MSNs have significant potential to serve as nanocarriers for pulmonary drug delivery [48]. Additionally, researchers need to carefully evaluate the safety and toxicity of nanoparticle-based drug delivery systems. While some studies have shown promising results, others have raised concerns about the potential for long-term toxicity and negative environmental impacts of nanoparticle-based drug delivery [49,50]. Currently, it is widely believed that the cytotoxicity of nanoparticles is mainly related to their large surface area and small size [51]. Yuan et al. concluded through their study on the effects of 20, 30, and 40 nm zinc oxide nanoparticles on human embryonic lung fibroblasts that cytotoxicity is concentration-dependent, therefore calling for the minimum therapeutic concentration [52]. Other researchers found that the surface charge and solubility are also associated with the cytotoxicity of nanoparticles [53,54].

Moving forward, researchers are exploring several future directions for nanoparticle-based drug delivery systems. For example, considering that there is a large amount of mucus oozing out of the lungs during chronic inflammatory diseases, researchers are developing new mucus-penetrating nanoparticles (MPPs). Uptake mechanism studies revealed that caveolae-mediated endocytosis and macropinocytosis contributed to the absorption of MPPs [55]. In vivo research results showed a more than five-fold increase in drug bioavailability [56]. Others are investigating new methods for optimizing nanoparticle design and surface modification to improve targeting and drug release [40,57]. Additionally, some researchers are investigating the potential of combining nanoparticles with other treatment modalities such as gene therapy or immunotherapy [46,58]. Finally, there is growing interest in developing personalized nanoparticle-based drug delivery approaches that can be tailored to individual patients based on their unique disease characteristics and genetic profiles [59].

Through targeted drug delivery, nanoparticles have the potential to improve therapeutic efficacy and reduce systemic side effects. Overall, nanoparticle-based drug delivery systems hold great promise for the treatment of chronic inflammatory respiratory diseases.

## 3. Inhaled Corticosteroids (ICSs)

Inhaled corticosteroids (ICSs) are widely used as a treatment option for chronic respiratory diseases such as asthma and chronic obstructive pulmonary disease (COPD). These medications work by reducing the production of inflammatory mediators in the airways, which helps prevent or reduce inflammation, bronchoconstriction, and mucus production. According to the Global Initiative for Asthma (GINA) report [1], ICSs have been shown to improve lung function, reduce exacerbation, and improve quality of life in patients with chronic respiratory diseases.

However, there are some current challenges with ICS delivery that limit their efficacy. One major challenge is achieving the optimal distribution of the medication throughout the lungs. ICS particles can become trapped in the mouth or throat, reducing their effectiveness in the lower airways [60]. Patients may also have difficulty using their inhaler correctly, leading to reduced medication delivery and efficacy [61]. Moreover, selecting the appropriate ICS dose for each patient can be challenging, as individual needs can vary significantly [62].

To optimize ICS delivery and improve its efficacy, several methods have been developed. One approach involves the use of spacer devices, which help to slow down the speed of medication delivery and improve medication deposition in the lungs [63]. Another approach is the development of more efficient ICS formulations, such as fine-particle ICSs, which have shown improved efficacy compared with conventional ICS formulations [64]. Fine-particle ICSs have greater deposition in the small airways compared with conventional ICSs [65]. According to a meta-analysis, fine-particle ICSs have significantly higher odds of achieving asthma control [66]. The combination of ICSs and other drugs is also worth further optimization (Figure 2). Additionally, research advancements have explored smart inhalers that can monitor medication adherence and provide feedback to patients [67]. Nowadays, four kinds of inhalers (nebulizers, dry powder inhalers (DPIs), pressurized metered-dose inhalers (pMDIs), and soft mist inhalers (SMIs)) are widely used (Table 2). Recently, artificial intelligence (AI)-based intelligent inhalers have attracted much attention, as they can enable real-time regulation of inhalation plans. For example, intelligent dry powder inhalers (DPIs) constructed based on artificial neural networks (ANNs) have effectively improved the bioavailability of drugs [68], but additional data are still needed to train more advanced models to output better drug delivery plans [69].

While there have been notable advancements, it is important to acknowledge that there are still existing limitations concerning the use of ICSs that necessitate careful consideration and remediation. For example, some studies have suggested that long-term use of ICSs may increase the risk of pneumonia and cataracts [82,83]. Moreover, further research is needed to determine the optimal ICS dose and duration of treatment for individual patients [84].

Future directions for research in ICS delivery are focused on several areas. Personalized ICS dosing strategies based on individual patient characteristics and disease severity are being explored [85]. Investigations are currently underway to explore new ICS formulations that utilize innovative drug delivery technologies, including nanotechnology and microencapsulation [86].

Thus, ICSs remain an effective treatment option for chronic respiratory diseases, but proper delivery optimization is crucial to their efficacy and safety.

## 4. Novel Biologicals

Biologicals are a class of drugs that are produced using living cells or organisms and have revolutionized the treatment of many respiratory diseases such as asthma, chronic obstructive pulmonary disease (COPD), and idiopathic pulmonary fibrosis (IPF). Biologicals target specific proteins and immune cells involved in the inflammation and damage of the airways and lungs, offering a more precise and effective treatment option compared with traditional medications [87].

Research is ongoing to identify new biological targets for the treatment of respiratory diseases. For example, interleukin-33 (IL-33) is a protein that has been shown to promote allergic inflammation in asthma and may be a potential target for biologicals [88]. Other targets include prostaglandin D2 (PGD2) and its receptor, chemoattractant receptor-homologous molecule expressed on T-helper type-2 cells (CRTH2), which are involved in airway smooth muscle contraction and inflammation [89], and the protein periostin, which plays a role in lung tissue remodeling in asthma and IPF [90]. These novel targets offer the potential for more personalized and targeted therapies for respiratory diseases (Table 3).

Effective delivery of biologicals to the lungs is critical for their efficacy. Various innovative methods have been developed to improve drug delivery, including nebulizers, dry powder inhalers, and intravenous infusions [104]. Additionally, recent advancements in nanotechnology have opened up new possibilities for targeted drug delivery to specific areas of the lungs [105]. For example, a new exosome membrane–modified M2 macrophages targeted nanomedicine has been proved to be effective for allergic asthma in vivo [40]. The progress of these delivery methods provides the potential for achieving the specific action of biopharmaceuticals at the organ level.

Research on biologicals for respiratory diseases has made significant advancements in recent years. For example, studies have shown the efficacy of biologicals targeting interleukin-5 (IL-5) and interleukin-4/13 (IL-4/13) in asthma [106] and the effectiveness of nintedanib, a tyrosine kinase inhibitor, in slowing the progression of IPF [107]. However, there are also limitations to biological therapy, including high costs and the risk of adverse effects such as allergic reactions and infections [108].

Developing personalized biological therapies and improving drug delivery methods will undoubtedly be the main trends in the future. For example, studies have explored the use of biomarkers to identify patients who may benefit from specific biologicals and the development of smart inhalers that can monitor adherence and provide feedback to patients [109]. Additionally, research is ongoing to develop new biologicals that target novel pathways and cells involved in respiratory diseases [110].

So far, biologicals have transformed the treatment of respiratory diseases, offering more precise and targeted therapies. Three anti-IL-5 biologicals and one anti-IL-4R biological have recently emerged as promising treatments for type 2 (T2) asthma [111]. There is also evidence that itepekimab could reduce the annualized exacerbation rate and improve lung function in former smokers with COPD [112]. Further research is needed to optimize the efficacy, safety, and cost-effectiveness of these treatments.

## 5. Gene Therapy

Gene therapy is a promising approach for the treatment of respiratory diseases, including asthma, cystic fibrosis, alpha-1 antitrypsin deficiency, and pulmonary hypertension. This therapeutic approach involves the delivery of genetic material to replace or supplement faulty genes, prevent the expression of harmful genes, or introduce new genes to cells [113]. Gene therapy offers the potential for long-lasting effects compared with traditional pharmacological treatments.

Several gene therapy approaches have been developed for respiratory diseases, including viral-vector-based delivery systems and clustered regularly interspaced short palindromic repeats–CRISPR-associated protein 9 (CRISPR–Cas9) technology. Viral vectors, such as adeno-associated viruses (AAVs) and lentiviruses, are commonly used to deliver the therapeutic gene to target cells. AAVs have shown promise in clinical trials for cystic fibrosis and other genetic lung diseases [114]. CRISPR–Cas9 technology allows precise editing of defective genes in living cells and has been used to correct mutations in animal models of cystic fibrosis and alpha-1 antitrypsin deficiency [115] (Table 4). However, there are still limitations to these approaches, such as immune responses to viral vectors and potential off-target effects of genome editing.

In recent years, research on gene therapy for respiratory diseases has achieved remarkable advancements. For example, clinical trials of AAV gene therapy targeting *CFTR* for cystic fibrosis have shown significant improvements in the lung function and quality of life in patients [124]. Additionally, promising results have been seen in preclinical studies using CRISPR–Cas9 gene editing for cystic fibrosis and other respiratory diseases [125]. However, it is important to acknowledge that there are still problems in this field, including the necessity for enhanced delivery techniques and thorough investigation of the potential risks linked to genome editing. Further research and development are imperative to answer these questions [126].

Gene therapy approaches for respiratory diseases remain to be optimized. This includes the development of more efficient and targeted delivery methods such as aerosolized nanoparticles for lung-specific delivery [127]. The natural wrapping property of exosomes can protect genetic material from degradation and attack by the immune system, making it an excellent carrier [128]. Additionally, research is exploring the use of gene therapy in combination with other therapies, such as stem cell therapy, to enhance therapeutic efficacy [129]. Furthermore, ethical considerations surrounding genome editing, including potential unintended effects and the need for informed consent, require continued discussion and investigation.

Gene therapy offers the potential for long-lasting effects compared with traditional pharmacological treatments. Ongoing research is needed to optimize the safety and efficacy of gene therapy approaches and to address the limitations and ethical concerns associated with this promising therapeutic approach.

## 6. Personalized Medicine

Personalized medicine is an approach to healthcare that considers individual variability in genes, environment, and lifestyle for the prevention, diagnosis, and treatment of diseases. In the context of respiratory diseases, personalized medicine aims to tailor treatment strategies to the unique needs of patients based on their genetic and molecular characteristics as well as other clinical and environmental factors [8]. Implementing this approach has the power not only to enhance patient outcomes but also to alleviate healthcare costs.

Personalized medicine has several advantages for patients with chronic inflammatory respiratory diseases such as asthma and chronic obstructive pulmonary disease (COPD). By identifying biomarkers and other factors (like serum immunoglobulins, sputum microbiome, and prognostic imaging biomarkers) that contribute to disease progression and exacerbation, physicians can develop more targeted treatment plans that minimize side effects and maximize efficacy [130]. For example, some patients with severe asthma may benefit from biological therapies targeting specific cytokines or immune cells. Additionally, personalized medicine may enable early identification of patients at risk for disease progression or exacerbation, allowing proactive interventions to prevent severe symptoms and hospitalizations.

Current research in personalized medicine for respiratory diseases is focused on identifying biomarkers and developing diagnostic tools to better classify patients based on their underlying disease mechanisms. For example, studies have identified gene expression profiles associated with different subtypes of asthma and COPD [131,132]. Additionally, researchers are exploring the use of wearable sensors and other technologies to monitor patient symptoms and disease activity in real time, enabling more timely interventions and adjustments to treatment plans.

The latest advancements in personalized medicine for respiratory diseases have exhibited promising outcomes, showcasing improved patient well-being and the capacity for cost savings within the healthcare system. For example, a study of biomarker-guided asthma management found significant reductions in asthma exacerbation and healthcare utilization compared with standard care [133]. However, there are still limitations to the implementation of personalized medicine in clinical practice, such as the cost and availability of diagnostic tests and therapies, as well as ethical considerations surrounding the use of genetic information in treatment decisions [134].

Improving the accuracy and accessibility of diagnostic tests and expanding the range of targeted therapies available to patients are key points of personalized medicine for respiratory diseases. For example, researchers are exploring the use of artificial intelligence and machine learning algorithms to better predict patient outcomes and identify optimal treatment strategies [135]. Additionally, studies are investigating the potential benefits of combining multiple targeted therapies for patients with complex disease mechanisms. Furthermore, ongoing discussions around ethical and regulatory issues will continue to shape the development and implementation of personalized medicine in clinical practice.

In conclusion, personalized medicine allows treatment plans to be more targeted, effective, and tailored to individual patient needs. Ongoing research is needed to address the limitations and ethical considerations associated with this approach and to optimize the accuracy and accessibility of diagnostic tests and therapies. Due to its high heterogeneity, personalized healthcare needs to be organically combined with various other therapies to revitalize the lungs of patients with chronic inflammatory diseases (Figure 3).

## 7. Conclusions

Targeted drug delivery systems, including nanoparticle-based systems, ICSs, novel biologicals, gene therapy, and personalized medicine, hold great promise for the treatment of chronic inflammatory respiratory diseases. Ongoing research focuses on developing new delivery systems that can specifically target lung cells while minimizing systemic side effects. Furthermore, novel biological targets and innovative methods for delivering biologicals to the lungs are also being explored. Gene therapy approaches, including viral-vector-based delivery systems and CRISPR–Cas9 technology, show potential for treating respiratory diseases. Personalized medicine approaches could improve treatment outcomes by tailoring therapies to individuals based on their unique characteristics. Finally, combining different drug delivery systems, such as using organ-specific nanoparticles to deliver gene-targeting drugs according to disease subtypes, can further enhance drug efficacy. The utilization of an exosome-based vector system, which efficiently and specifically delivers mRNA or CRISPR–Cas9 plasmids to target cells, also holds promise for targeted gene therapy both in vitro and in vivo.

The clinical implications of these advancements are significant, as targeted drug delivery systems have the potential to improve patient outcomes and reduce healthcare costs. Healthcare professionals should consider integrating these approaches into their practice as they become more widely available. However, there is still a considerable journey from the laboratory bench to clinical application. Hence, additional research is needed to refine and optimize these approaches for maximum effectiveness. It is important to address safety concerns related to nanoparticle-based delivery systems and gene therapy as well as to develop improved methods for delivering biologicals to the lungs. Moreover, identifying optimal personalized medicine approaches is of paramount importance to ensure that treatments align with the specific demands and characteristics of individual patients.

To sum up, the use of targeted drug delivery systems represents a promising approach to the treatment of chronic inflammatory respiratory diseases. Further research is required to fine-tune and optimize these approaches as well as to identify the most effective personalized medicine strategies. For instance, utilizing biological models that closely resemble the human lung environment, such as lung organoids, can better reflect the effect of new drug delivery systems. Given the clinical heterogeneity of chronic inflammatory pulmonary disease, machine learning methods offer distinct advantages in calculating personalized treatment plans and predicting treatment outcomes in advance, leveraging the patient’s phenotype, subphenotype, and internal characteristics. Furthermore, defining refined subtypes of chronic inflammatory lung diseases based on multiple omics features can better capture the unique characteristics of each patient. Ultimately, the goal is to improve patient outcomes and reduce healthcare costs by delivering treatments that are tailored to individual patient needs.

## Figures and Tables

**Figure 1 pharmaceutics-15-02151-f001:**
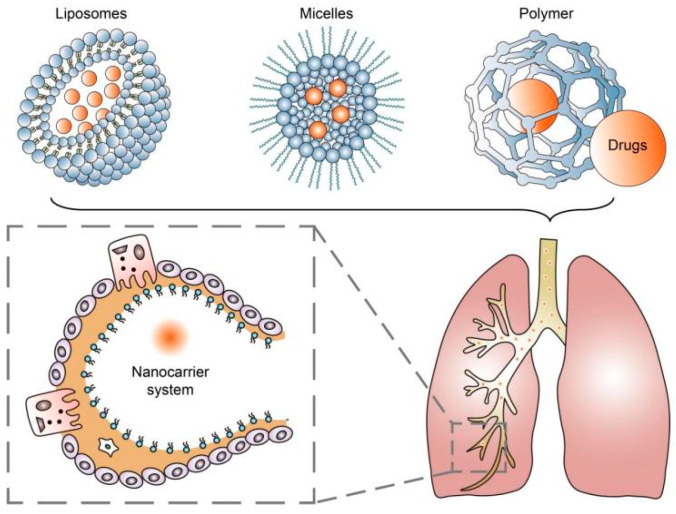
Nanocarrier systems can achieve drug deposition and controlled release in the lungs.

**Figure 2 pharmaceutics-15-02151-f002:**
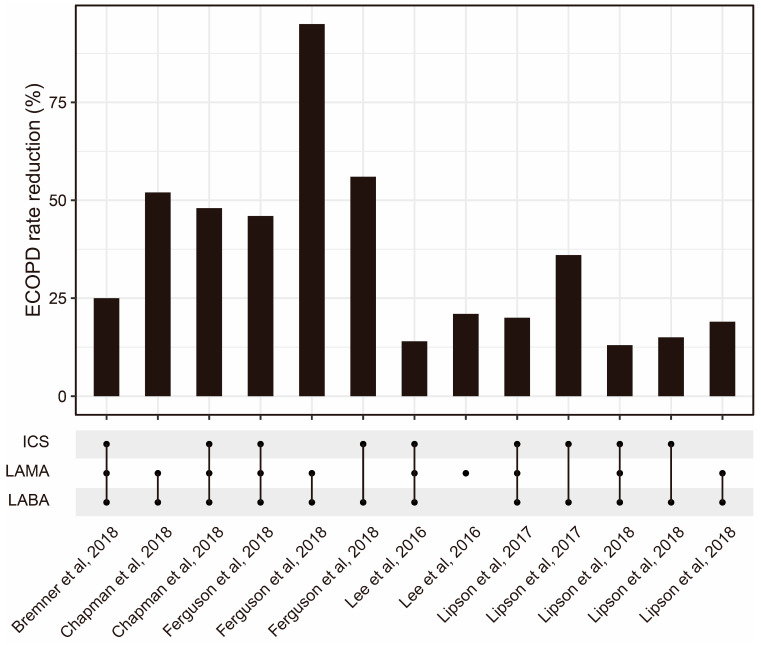
The ECOPD rate reduction from ICSs combined with other drug regimens reported by some published studies [70,71,72,73,74,75]. Abbreviations: ECOPD: Exacerbation of chronic obstructive pulmonary disease; ICS: Inhaled corticosteroid; LAMA: Long-acting muscarinic antagonist; LABA: Long-acting beta2-adrenergic agonist.

**Figure 3 pharmaceutics-15-02151-f003:**
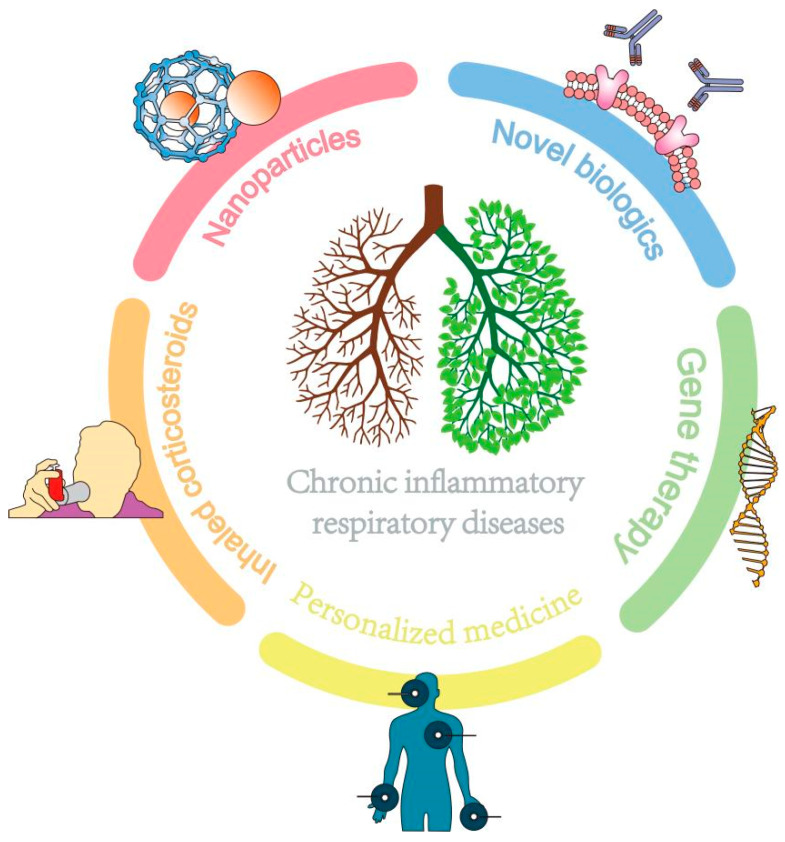
The combination of multiple therapies benefits patients with chronic inflammatory lung diseases.

**Table 1 pharmaceutics-15-02151-t001:** Therapeutic applications of nanoparticles in chronic inflammatory respiratory diseases.

Diseases	Type of Nanoparticles	Drugs	Target Ligands	Targets	References
Asthma	SPION	None	IL4Rα monoclonal antibody	ASMs	[35]
	SPION	None	Anti-ST2 blocking antibodies	Inflammatory lung tissue	[36]
	PLGA-based nanoparticles	Smart silencer of Dnmt3aos	Exosome membrane of M2 macrophages	M2 macrophages	[40]
	LNP	Polyinosinic-polycytidylic acid	None	Lung epithelial cells	[41]
COPD	HNP	siRNA against SCNN1A and SCNN1B	None	Lung epithelial cells	[42]
	LNP	siRNA against TNF-α	None	None	[43]
IPF	LNP	siRNA against IL-11	None	MLFs	[44]
CF	LNP	Plasmid DNA	ICAM-1 targeting peptide	Lung epithelial cells	[45]

Abbreviations: SPION: Superparamagnetic iron oxide nanoparticle; IL4Rα: Interleukin-4 receptor alpha; ASM: Airway smooth muscle cell; ST2: Grow stimulation expressed gene 2; PLGA: Polylactic-co-glycolic acid; LNP: Lipid nanoparticle; HNP: Hybrid nanoparticle; SCNN1A: Sodium channel non-alpha subunit 1A; SCNN1B: Sodium channel non-alpha subunit 1B; TNF-α: Tumor necrosis factor alpha; IL-11: Interleukin-11; MLFs: Mouse lymphatic fibroblasts; ICAM-1: Intercellular adhesion molecule-1.

**Table 2 pharmaceutics-15-02151-t002:** Different kinds of ICS inhalers.

Type of Inhaler	Subtype	Characteristics	Advantages	Limitations	References
Nebulizers	Jet (or pneumatic)	Use compressed air or oxygen to convert liquid medication into a fine mist for inhalation.	Versatile and suitable for all ages.	Longer administration times, produce noise and vibration, require power sources, and need regular maintenance.	[76]
	Ultrasonic	Use high-frequency vibrations to convert liquid medication into a fine mist for inhalation.	Portable and compact, have faster administration times, operate quietly.	Not suitable for medications that are heat-sensitive or contain suspensions.	[77]
	Mesh	Use a vibrating mesh or perforated plate to generate a fine aerosol mist from liquid medication.	Portable, lightweight, and operate silently with faster administration times.	Have limitations in delivering higher viscosity medications or large medication volumes.	[78]
Dry powder inhalers (DPIs)	Single- and multi-unit doses	Deliver medication directly to the lungs in a powdered form.	Breath-activated, portable, and do not require coordination between inhalation and device activation.	Require adequate inspiratory flow for optimal drug delivery, and can be used only with specific types of dry powder medications.	[79]
Pressurized metered-dose inhalers (pMDIs)	Single and combined drugs	Deliver medication in a pressurized aerosol form using propellants.	Deliver a consistent dose, require minimal preparation time.	The presence of propellants and the inability to assess remaining medication levels easily.	[80]
Soft mist inhalers (SMIs)	None	Deliver medication as a slow-moving aerosol mist.	Provide consistent and precise dosing, generate a slow-moving mist suitable for patients with diverse inspiratory abilities, and are equipped with dose counters to monitor medication levels.	Potential clogging if not used properly, higher cost compared with other inhalers, and limited availability of medications in soft mist formulation.	[81]

**Table 3 pharmaceutics-15-02151-t003:** Novel biological targets of chronic inflammatory respiratory diseases.

Chronic Inflammatory Respiratory Diseases	Targets	Mechanism/Effect/Receptor	Research Progress	References
Asthma	IL-33	When IL-33 binds to its receptor ST2, it can trigger inflammation and airway hyperresponsiveness.	The knockdown of P2Y_13_-R can regulate the release of IL-33 and prevent experimental asthma.	[88]
	TSLP	A cytokine involved in regulating the immune system.	TSLP can promote the activation of ILC2 and induce congenital allergic inflammation.	[91]
	CRTH2	A receptor mainly expressed on Th2 cells.	A CRTH2 antagonist (OC000459) can effectively reduce the increase in eosinophils and swelling of nasal mucosa. CRTH2 and TP antagonists have been registered for clinical use in asthma.	[92]
	IL-4/IL-13	Two cytokines involved in regulating immune responses.	IL-4/IL-13 stimulate CCL-11 production to alleviate HDM-induced asthma.	[93]
	IL-25	Also known as IL-17E, a cytokine primarily expressed in respiratory epithelial cells.	IL-25 induces excessive production of ROS through AMPK-related mitochondrial autophagy, leading to airway inflammation and remodeling in asthma.	[94]
	OX40	A co-stimulatory molecule that plays an important role in T-cell activation.	OX40-deficient mice exhibit reduced lung inflammation and weakened airway hyperresponsiveness	[95]
	S1P	A physiologically active lipid molecule that plays an important role in the immune system.	Bronchial specimens harvested from S1P-overexpressing mice showed overexpression of EMT-related markers and bronchial hyperresponsiveness.	[96]
IPF	TGF-β	An important growth factor that plays an important role in the pathological process of IPF.	Nestin knockdown inhibited TGF-β signaling by suppressing the recycling of TβRI to the cell surface.	[97]
	IL-11	IL-11 activates multiple signal transduction pathways by binding to its receptor, IL-11Rα, thereby promoting the activation and proliferation of fibroblasts.	An inhalable and mucus-penetrative nanoparticle (NP) system incorporating siRNA against IL11 (siIL11@PPGC NPs) hindered fibroblast differentiation and reduced ECM deposition via inhibition of ERK and SMAD2.	[44]
	PDGF	A cell growth factor that is involved in fibrocyte proliferation, inflammatory response, and the occurrence of IPF.	Nintedanib, a potent small-molecule inhibitor of the receptor tyrosine kinases PDGF receptor, has shown consistent anti-fibrotic and anti-inflammatory activity in animal models of lung fibrosis.	[98]
	Wnt/β-catenin	An important signaling pathway involved in biological processes such as cell proliferation and differentiation.	Activation of Wnt/β-catenin led to a significant increase in IL-1β and IL-6 in mice.	[99]
	MMPs	MMPs are involved in the process of lung tissue remodeling and fibrosis.	Clinical research reports show a significant increase in MMP levels in blood and lung samples from patients with IPF. Most MMPs can promote the development of IPF in mouse models.	[100]
CF	CFTR	The CFTR protein forms a channel on the cell membrane that primarily regulates chloride ion (Cl^−^) transport, maintaining water and salt balance. When the CFTR gene mutates, it can affect the production and excretion of mucus.	Currently available CFTR modulators: ivacaftor, lumacaftor, Orkambi (a combination of lumacaftor and ivacaftor).	[101,102,103]

Abbreviations: IL-33: Interleukin-33; ST2: Grow stimulation expressed gene 2; P2Y_13_-R: P2Y purinoceptor 13 receptor; TSLP: Thymic stromal lymphopoietin; ILC2: Type 2 innate lymphoid cell; CRTH2: Chemoattractant receptor-homologous molecule expressed on T-helper type-2 cells; TP: thromboxane receptor; IL-4/IL-13: Interleukin-4/interleukin-13; CCL-11: C-C motif chemokine ligand 11; HDM: House dust mite; IL-25: Interleukin-25; IL-17E: Interleukin-17E; ROS: Reactive oxygen species; AMPK: Adenosine 5′-monophosphate (AMP)-activated protein kinase; OX40 (Tnfrsf4): Tumor necrosis factor receptor superfamily, member 4; S1P: Sphingosine-1-phosphate; EMT: Epithelial–mesenchymal transition; TGF-β: Transforming growth factor-beta; TβRI: Transforming growth factor-beta receptor I; IL-11: Interleukin-11; IL-11Rα: Interleukin-11 receptor alpha; ECM: Extracellular matrix; ERK: Extracellular signal-regulated kinase; SMAD2: Small mothers against decapentaplegic family member 2; PDGF: Platelet-derived growth factor; IL-1β: Interleukin-1β; IL-6: Interleukin-6; MMPs: Matrix metalloproteinases; CF: Cystic fibrosis; CFTR: Cystic fibrosis transmembrane conductance regulator.

**Table 4 pharmaceutics-15-02151-t004:** Treatments based on gene therapy for chronic inflammatory respiratory diseases.

Chronic Inflammatory Diseases	Gene	Research Progress	References
Asthma	*IL-12*	Overexpression of single chain *IL-12* (scIL-12) through rAAV vector significantly suppressed the total number of cells and eosinophil infiltration as well as the mucus secretion in mice.	[116]
	*CTNNAL1*	Airway hyperresponsiveness and inflammation were significantly attenuated in mice pretransduced with AAV-miR-511-3p.	[117]
	*MIF*	Intratracheal adeno-associated virus (AAV) vector (*MIF*-mutant AAV57) could reduce airway remodeling in asthmatic mice.	[118]
	SNP rs12946510	CRISPR–Cas9 genome editing demonstrated that the SNP of rs12946510 was associated with asthma.	[119]
AATD (An important cause of COPD)	*SERPINA1*	The systemic delivery of AAV8-CRISPR, targeting exon 2 of h*SERPINA1*, and the AAV, which provided the donor template to correct the Z mutation, could both restore modest levels of wildtype AAT-M in a mouse model of AATD.	[120]
		rAAV-mediated *SERPINA1* gene augmentation largely preserved lung tissue elasticity and alveolar wall integrity in mice models.	[121]
		The therapeutic application of CRISPR–Cas9 for genome editing in a humanized mouse model successfully mitigated the phenotype of AATD.	[122]
CF	*CFTR*	Complementing *CFTR* in CF pigs with AAV rescued the anion transport defect in a large-animal CF model.	[123]

Abbreviations: *IL-12*: Interleukin-12; rAAV: Recombinant adeno-associated virus; *CTNNAL1*: Catenin alpha-like 1; *MIF*: Macrophage migration inhibitory factor; SNP: Single nucleotide polymorphism; CRISPR–Cas9: Clustered regularly interspaced short palindromic repeats/CRISPR-associated protein 9; AATD: Alpha-1-antitrypsin deficiency; *SERPINA1*: Serpin family A member 1; CF: Cystic fibrosis; *CFTR*: Cystic fibrosis transmembrane conductance regulator.

## Data Availability

Not applicable.

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
