# Peer review of "Advancing Treatment Strategies: A Comprehensive Review of Drug Delivery Innovations for Chronic Inflammatory Respiratory Diseases"

_pharmaceutics, 2023, doi:10.3390/pharmaceutics15082151_

Round 1

Reviewer 1 Report

This  review focuses on the role of drug delivery systems in chronic inflammatory respiratory diseases, particularly nanoparticle-based drug delivery systems, inhaled corticosteroids (ICS), novel biologics, gene therapy, and personalized medicine.

It is well written and comprehensive,anyway  i would like to suggest some small modifications.

In my opinion considering the nanoparticle-based delivery system section some indication of how the nanoparticles can be made mucopenetrating should be described; moreover the magnetofection technique should be a little explicited.

I wonder if there are novel biological targets for reducing inflammation also in cystic fibrosis or not, in order to insert them in table 2 together with asthma and IPF.

I noticed that the name of the last author is missing, please fix it.

Author Response

Dear Reviewer,

Thank you for your valuable feedback on our review. We appreciate your positive remarks regarding the comprehensive coverage of the topic. We have carefully considered your suggestions and have made the necessary modifications to address them.

  1. Regarding the section on nanoparticle-based delivery systems, we have now included a brief description of how nanoparticles can be made mucopenetrating(Line 153-158). Additionally, we have provided further explanation on the magnetofection technique to enhance clarity and understanding (Line 108-116).

  1. In relation to your query about novel biological targets for reducing inflammation in cystic fibrosis, we have thoroughly researched this area and found promising targets that could be included in Table 3 (Line 250).

  1. Lastly, we apologize for the oversight in omitting the last author's name. We have rectified this error and included the complete authorship information(Line 4).

Once again, we sincerely appreciate your insightful comments and assure you that we have addressed all the issues you raised (They are highlighted in green in the manuscript. Please see the attachment). Thank you for your time and consideration.

Best regards,

Junming Wang

Reviewer 2 Report

It is a nice compilation of information on innovative drug delivery systems for chronic inflammatory respiratory diseases. Authors should also focus on statistics, etiology, pathogenesis and symptoms of chronic inflammatory respiratory diseases in short. I will recommend accepting the review with some major changes which are mentioned in the form of comments in pdf attached. 

Minor editing of English language required

Author Response

Dear Reviewer,

Thank you for your valuable feedback on our review article titled "Advancing Treatment Strategies: A Comprehensive Review of Drug Delivery Innovations for Chronic Inflammatory Respiratory Diseases." We truly appreciate your insightful comments and suggestions, which have greatly improved the overall quality of our manuscript.

Regarding your suggestion to include statistics, etiology, pathogenesis, and symptoms of chronic inflammatory respiratory diseases, we have carefully revised the manuscript to address this concern. We have incorporated a succinct overview of these aspects, providing a comprehensive background for readers to better understand the context and significance of innovative drug delivery systems in managing these diseases.

We have also thoroughly reviewed the attached PDF document containing your specific comments and have made the necessary changes accordingly (They are highlighted in yellow in the manuscript). We believe that these modifications have significantly strengthened the scientific content and clarity of our article.

  1. We’ve consulted a large number of literature and provide the latest statistic and etiology of asthma and COPD in short(Line 34-37).

  1. After reviewing the existing research results, we found that the targets of the new nanoparticle-baseddrug delivery system are mainly lung epithelial cells and macrophages (Line 46).

  1. We have addressed your suggestion by incorporating the Table 1 (Line 125)summarizing the current research findings on nanoparticles-based drug delivery systems. We believe that the inclusion of such a table greatly enhances the readability and accessibility of the manuscript, allowing readers to grasp the key findings from various studies in a concise format. We have also rewrite the “Nanoparticle-based drug delivery systems” section to present specific information regarding the type of nanoparticles, their release pattern, target, in vitro or in vivo findings, and significance. It provides a comprehensive overview of the research articles in this field (Line 90-123)

  1. We briefly introduced the great plasticity and controllability of MSN to support its significance as a nanocarrier material(Line 135-141).

  1. The advantages of fine-grained ICS compared to traditional inhaled ICS and its proven value in clinical research are briefly introduced(Line 193-195).

  1. We have collected relevant literature and were pleasantly surprised to find that there have been many studies on AI based intelligent inhalers. In this article, we have reviewed them and roughly introducedANN based intelligent inhalers (Line 200-205).

Once again, we express our gratitude for your time and effort in reviewing our work. We are pleased to inform you that all the major revisions you proposed have been implemented. We hope that the revised manuscript now meets your expectations and that you will recommend its acceptance for publication.

Please let us know if there are any further suggestions or concerns. We look forward to your response.

Sincerely,

Junming Wang

Reviewer 3 Report

This manuscript reviews innovative drug delivery strategies for inflammatory respiratory diseases.  This manuscript aims to review nanoparticle-based delivery systems, inhaled corticosteroids, novel biologicals, gene therapy and personalized medicine with an emphasis on advances and potential for use in the treatment of inflammatory diseases of the airways.  In general, the manuscript is well written, concise and to the point.  This manuscript will be of value and relevant for researchers and scientists working in the field of inflammatory diseases of the airways. 

Minor corrections and suggestions to improve the quality of the manuscript have been indicated in the attached review copy of the manuscript using the “sticky notes”.  Furthermore, the following aspects of the manuscript should also be noted: 

1.      In line 112-114: Thew following statement is made: “While some studies have shown promising results, others have raised concerns about the potential for long-term toxicity and negative environmental impacts of nanoparticle-based drug delivery”. The authors need to elaborate here as it is a review manuscript.  For instance, it is well-known that there are toxicity concerns as well as targeting issues with nanoparticle-based delivery systems due to their small size.   

2.       Starting in line 186, in the section on “New biologics”, the authors, use "biologics" and "biologicals" interchangeably.  Use the either the term biologicals or biologics throughout the manuscript for consistency, please. 

3.      Line 204, Table 2, column 3, the heading of the column is Mechanism/effect.  Consider, including the word receptor in the heading as this column does not always refer to a  mechanism/effect but sometimes only refer to a receptor or revise the different lines to include an effect or mechanism.   

4.      Line 237:  the authors state: “So far, biologic drugs have transformed the treatment of respiratory diseases, of more precise and targeted therapies.”  How did biological drugs transform the treatment of respiratory disease?  This is an important conclusion. The authors need to substantiate this statement with evidence such as clinical evidence and examples of biologicals registered for treatment of airway diseases with regulatory authorities. 

Author Response

Dear Reviewer,

Thank you for your thorough review of our manuscript titled "Advancing Treatment Strategies: A Comprehensive Review of Drug Delivery Innovations for Chronic Inflammatory Respiratory Diseases". We appreciate your positive feedback and valuable suggestions, and we have made the necessary revisions to address each point (They are highlighted in blue in the manuscript. Please see the attachment).

  1. We have elaborated on the concerns regarding nanoparticle-based drug delivery systems. Additional information on the potential long-term toxicity and environmental impacts associated with nanoparticlesis now included in the manuscript. Specifically, we discussed possible factors related to the cytotoxicity caused by nanoparticles (Line 142-150).

  1. We have revised the manuscript to consistently use the term "biologicals"(Line 232).

  1. We have revised the heading of column 3 of Table 3to "Mechanism/Effect/Receptor" to accurately reflect the content of the column. This change ensures that readers understand that the column may refer to either a mechanism, effect, or receptor (Line 250).

  1. We have provided further evidence to substantiate the statement that biologicaldrugs have transformed the treatment of respiratory diseases. We now include examples of registered biologicals for the treatment of asthma and COPD and refer to clinical evidence supporting their efficacy. This revised section highlights the significant impact of biological drugs in revolutionizing respiratory disease treatment (Line 285-290).

  1. Thank you so much for your meticulous attention to detail and for pointing out the grammatical errors in my manuscriptand imperfection in Figure 1. I sincerely appreciate your assistance, and I am pleased to inform you that I have made all the necessary corrections (Line 37, 47, 88, 180, 196, 200 and 263).

Once again, we sincerely appreciate your valuable input and constructive feedback. We believe that the manuscript has been significantly improved as a result of your suggestions. If you have any further comments or concerns, please do not hesitate to let us know.

Thank you for your time and consideration.

Best regards,

Junming Wang

Round 2

Reviewer 2 Report

All the necessary suggestions and changes have been implemented.